# Recent Advances in Cell Membrane Coated-Nanoparticles as Drug Delivery Systems for Tackling Urological Diseases

**DOI:** 10.3390/pharmaceutics15071899

**Published:** 2023-07-06

**Authors:** Cenchao Yao, Dahong Zhang, Heng Wang, Pu Zhang

**Affiliations:** 1Urology & Nephrology Center, Department of Urology, Zhejiang Provincial People’s Hospital, Affiliated People’s Hospital, Hangzhou Medical College, Hangzhou 310014, China; 17816542921@163.com (C.Y.); zppurology@163.com (D.Z.); 2The Second Clinical Medical College, Zhejiang Chinese Medical University, Hangzhou 310053, China

**Keywords:** CMNPs, urological disease, drug delivery

## Abstract

Recent studies have revealed the functional roles of cell membrane coated-nanoparticles (CMNPs) in tackling urological diseases, including cancers, inflammation, and acute kidney injury. Cells are a fundamental part of pathology to regulate nearly all urological diseases, and, therefore, naturally derived cell membranes inherit the functional role to enhance the biopharmaceutical performance of their encapsulated nanoparticles on drug delivery. In this review, methods for CMNP synthesis and surface engineering are summarized. The application of different types of CMNPs for tackling urological diseases is updated, including cancer cell membrane, stem cell membrane, immune cell membrane, erythrocytes cell membranes, and extracellular vesicles, and their potential for clinical use is discussed.

## 1. Introduction

Urological diseases encompass many complex disorders, including tumors, inflammation, infection, acute kidney injury, etc., and have become a global public health problem that collectively affects more than 1 in 10 individuals worldwide, resulting in high healthcare costs [1,2,3,4]. Moreover, it tends to develop at a younger age [5]. Current strategies for tackling urological diseases are ineffective due to physiological barriers in urological systems [6,7]. The application of nanomedicine is becoming more widespread [8] in urological disease treatment because it improves the pharmaceutical performance of drugs. Nevertheless, nanoparticles will be recognized and fast removed by the reticuloendothelial system (RES) as foreign elements in the body, causing immune responses and toxic effects, which will be the major hurdles that almost all platforms must overcome [9,10]. Based on recent research findings, it has been observed that over 85% of the re-ported stealth nanomaterials experience a rapid decline in blood concentration, reducing to half of the administered dose within 1 h after administration. This indicates that the nanoparticles have not achieved a satisfactory stealth effect [11]. PEGylation is a widely employed technique for prolonging blood circulation. However, membrane coating aligns more closely with the concept of holism due to the intricate structural hierarchy of biological systems.

Cell membrane coated-nanoparticles (CMNPs) form a biomimetic structure that inherits the surface properties and functions of the source cells and imparts additional biological capabilities to the encapsulated nanoparticles, such as immune evasion, cell-specific targeting, and extension of the systemic circulation [12,13,14,15,16]. Meanwhile, natural cell membranes will preserve the integrity and bioactivity of nanoparticles in vivo. Recent years have witnessed the increasing interest in the use of CMNPs as drug delivery systems to treat urological diseases, which is rarely reviewed elsewhere [17]. In this review, the current state of nanomedicine for treating urological diseases is briefly summarized, and the methods for CMNPs synthesis and surface engineering are updated. The application of different types of CMNPs for tackling urological diseases, especially tumors, inflammation, and acute kidney injury, is reviewed.

## 2. Current State of Nanomedicine for Treating Urological Diseases

According to the etiology, urological diseases are mainly classified into urological cancers, stones, infections, inflammation, and injuries [18]. The incidence of these diseases is increasing annually, leading to significant social burdens and high healthcare costs. Although surgery is the primary treatment for urological cancers, stones, and injuries, its clinical outcomes often fall short of expectations. For example, 25% to 30% of patients with renal cancer present with distant metastatic disease at the time of diagnosis [19], and approximately 40% of patients who receive surgical treatment eventually develop recurrence [20]. For infections, inflammation, and the remaining part of urological diseases which do not need surgical intervention, drug treatment is most widely used, but not that effective. There remain obstacles to drugs reaching and remaining in their targeted lesions. (1) Inherent physiological permeability barriers impede the delivery of therapeutic agents to diseased tissues and cells, such as the urothelial barrier [21], glomerular filtration barriers (GFB) [22], and blood–prostate barrier (BPB) [23]. (2) Off-targeting occurs in normal tissues frequently, leading to a high risk of side effects [24]. (3) Another challenge is the lack of subcellular targeted therapy, via which a specific type of drug can directly interact with its subcellular target to maximize therapeutic effects [25]. (4) Short-term retention of drugs in their targeted lesions determines their short-acting effects.

Nanoparticles, composed of materials such as lipids, metals, synthetic and biopolymeric polymers, proteins, and inorganic and organometallic compounds, have shown promise in improving the in vivo performance of loaded drugs [26,27,28,29]. (1) Nanomaterials have been long pursued as drug delivery systems to conquer urological physiological barriers [30,31]. For example, surface modification of drugs with mucus-penetrating moieties or hydrophobic ligands could accelerate the translocation of drugs across the mucus barrier or the epithelial barrier, respectively [32,33,34,35]. (2) Nanomaterials can reduce the off-target effect of drugs via passive targeting or active targeting [36]. In the past, nanoparticles with a hydrodynamic diameter (HD) within the range of 100 to 400 nm were considered ideal for passive tumor targeting due to the enhanced permeability and retention (EPR) effect. However, a recent study conducted by Cabral et al. has highlighted that micelles measuring 30 nm in size could potentially achieve much more effective tumor penetration when compared to micelles with a size of 100 nm [37]. (3) Nanomaterials correct the intracellular trafficking of drugs into the expected subcellular compartments [38]. For instance, nuclear localization signals (NLS)-labeled nanoparticles could facilitate the active nuclear transport of drugs [39,40]. Nanoparticles can also enter mitochondria through a variety of pathways. In one study, Tritylphosphine (PPh3)-coupled nanoparticles could send doxorubicin to mitochondria [41]. The application of organelle-targeted nanoparticles has also expanded to drug delivery to the endoplasmic reticulum [42] and Golgi apparatus [43]. (4) The prolonged exposure of drugs to their targeted lesions can be achieved via the combined use of nanomaterials. Mucoadhesive nanomaterials could adhere to the mucus barrier to allow the sustained release of drugs [30]. At the same time, mechanisms also exist within urological diseases to facilitate the targeted delivery of nanoparticles. These mechanisms include the EPR effect [44], the presence of specific receptors on the surface of tumor cells [45], alterations in the tissue microenvironment [46], and the involvement of inflammatory factors.

Among all types of nanoparticles, CMNPs have gained significant attention as drug delivery vehicles. CMNPs exhibit a biomimetic structure that retains the surface characteristics of the source cells and imparts bioactivities to the encapsulated nanoparticles, including immune evasion, cell-specific targeting, and extension of systemic circulation [12,13]. Apart from these inherent functionalities, the lipid layers on the outer surface of CMNPs are easily amenable to surface engineering. Furthermore, CMNPs are highly biocompatible drug delivery systems. From a translational nanomedicine perspective, CMNPs offer a promising alternative.

## 3. Synthesis and Surface Engineering of CMNPs

Hu et al. [47] pioneered the study of CMNP fabrication in 2011. They applied red blood cell membranes to shroud poly (lactic-co-glycolic acid) (PLGA) nanoparticles and prolong the circulation time of the CMNPs up to 72 h. Building on their work, a general consensus has been established regarding the synthesis procedure of CMNPs, which can be broadly divided into three steps: extraction of cell membranes, fusion of cell membranes with core nanoparticles, and surface engineering of CMNPs.

### 3.1. Extraction of Cell Membrane

Cell lysis and membrane purification constitute two critical steps in the extraction of cell membranes [48]. It is important to note that the extraction procedures vary depending on whether the source cells are nucleus-free.

For nucleus-free cells such as erythrocytes and platelets, the cells are lysed through hypotonic treatment or repeated freeze-thaw cycles. Subsequently, the purified cell membrane is obtained by removing soluble cytoplasmic components using centrifugation [49]. In order to maintain the biological activity of membrane proteins, protease inhibitors should be added to the extracted cell membranes and stored at 4 °C. 

The extraction and purification of eukaryotic cell membranes are more complex compared to nucleus-free cells, mainly because the removal of cell nuclei is required [50]. First of all, similar to the extraction of nucleus-free cell membranes, a sufficient number of cells must be collected for concentrating and purifying cell membranes, which are disrupted by incubation in hypotonic lysate or repeated freeze-thaw treatments. Afterward, the nucleus and intracellular biomolecules were removed by discontinuous sucrose gradient centrifugation or differential centrifugation. The membrane-rich fraction obtained from the previous step is washed with a plasma buffer. To obtain cell membrane vesicles, the fraction can be further processed through sonication or extrusion using porous membranes [51].

### 3.2. Fusion of Cell Membranes with Core Nanoparticles

The most commonly used method of CMNPs fusion is physical extrusion [47], during which core nanoparticles and the extracted purified cell membranes are co-extruded through a porous membrane using a micro-extruder. The mechanical force of extrusion can disrupt the integrity of cell membranes and reorganizes them to surround core nanoparticles. Although this method is convenient to perform and effective in producing CMNPs, it is difficult to achieve large-scale production due to the material loss caused by material deposition onto filter membranes [52].

Another method, ultrasound treatment, utilizes ultrasound waves in the range of 20 to 50 kHz to disrupt the integrity of cell membranes. The attractive force between the nanoparticles and the cell membrane fragments facilitates the formation of cell membrane vesicles with a spherical shape, which then surround the core nanoparticles [53]. Ultrasound treatment is more efficient for scaled-up production of CMNPs, but it may not be suitable for certain nanoparticles as sonication can affect the size and stability of the core nanoparticles [54].

Under the stimuli of the external electric field, the electroporation can increase the semi-permeability of cell membranes and induce the formation of a large number of micropores on the cell membrane, through which nanoparticles can diffuse into the cell membrane [55]. The electroporation avoids the destruction of nanoparticles and is less time-consuming and labor-intensive nature than the co-extrusion method [55]. However, its operating procedure is more complex. Additionally, the electroporation method can be combined with microfluidic devices to promote the productivity of CMNPs. Rao et al. injected a mixture of RBC membrane-derived vesicles and Fe_3_O_4_ nanoparticles into a microfluidic device, where Fe_3_O_4_ magnetic nanoparticles (MNs) and RBC membrane vesicles flow through the electroporation zone and eventually fused with the assistance of electrical pulses [56]. As a result, microfluidic electroporation can produce CMNPs with uniform sizes. Its high reproducibility also guarantees the potential for the large scale-up production of CMNPs with enhanced colloidal stability and in vivo bioactivity [57].

In situ polymerization is a technique employed to coat isolated cell membranes onto nanocomposites, enabling the growth of nanoparticle cores inside cell membrane-derived vesicles. This process involves the use of a monomer, an initiator, and a cross-linker, which, when exposed to radiation or heat, initiates polymerization. In situ polymerization offers several advantages, including the assurance of coating integrity, simple encapsulation of the payload, and easy control over the size and stiffness of the resulting CMNPs [58]. Zhang et al. utilized RBC membrane-derived vesicles as nanoreactors to synthesize polymeric cores through in situ polymerization, resulting in cell membrane-coated hydrogel nanoparticles [59]. By selectively polymerizing monomers upon ultraviolet exposure, a stable structure is formed. This approach effectively eliminates potential risks of content leakage during the nano gel preparation process [60].

A comparison of different fusion methods is summarized in Table 1.

Noteworthy, the orientation of cell membrane coating on the surface of core nanoparticles regulates the bioactivity of CMNPs [61]. The right-side-out oriented cell membranes can inherit the biological function from their parent cells, therefore providing core nanoparticles with immune-evasive stealth and prolonging the in vivo circulation. The inside-out cell membranes expose binding domains to membrane-impermeable drugs, such as small-molecule tyrosine kinase inhibitors, and, thus, they act as sponges to adsorb these drugs [61]. During the fusion procedure, correctly orientated cell membrane coating cannot be achieved via random cell membrane-core pairs. A more reliable strategy relies on the specific interaction between ligands anchoring at the surface of core nanoparticles and the intracellular domain of their corresponding receptors [62], thus connecting the surface of core nanoparticles with the inner side of the cell membrane. By this principle, the right-side-out oriented assembly of CMNPs can be achieved via the cytoplasmic domain of band 3- the P4.2-derived peptide pairs [62] and inside-out oriented assembly of CMNPs can be achieved via azide analogues in the glycan-alkynyl pairs [63], biotin-streptavidin pairs [61], and sucrose density gradient centrifugation [64]. To verify the corrected assembly of CMNPs, the presence of two cell membrane markers at the outer surface of CMNPs is tested. CD47 is a transmembrane protein that contains a variable extracellular N-terminal domain, a presenilin domain with five transmembrane regions, and a C-terminal intracellular region [65]. Under the view of the transmission electron microscope (TEM), immunogold staining of different domains of CD47 can reveal the spatial relationship of different layers of cell membranes with core nanoparticles [66]. Considering that inside-out cell membrane patches and right-side-out cell membrane patches may assemble onto core nanoparticles together, immune electron microscopy is not effective in quantifying the exact proportion of differently oriented cell membrane patches on the surface of core nanoparticles. Sialic acid exclusively appears as a terminal residue of extracellular surface glycans [67]. Sialidase can dissociate the sialic acid from the outside surface of CMNPs, which can be further quantified via commercial kits [63]. The equivalent density of sialic acid from the outside surface of CMNPs to that from the outside surface of natural cells is solid evidence on the right-side-out cell membrane coating.

Loss of membrane integrity in CMNPs may cause drug leakage from core nanoparticles during drug delivery, unwanted biomolecule adsorption in physiological fluids, altered mechanical properties of nanoparticles, and changed molecules affinity of the membrane. The ratio of full cell membrane coated-nanoparticles in CMNPs is as low as 1.8 ± 0.1%, 6.2 ± 0.3%, and 6.5 ± 0.3% via sonication and extrusion [68]. To further improve the cell membrane integrity, red blood cell membrane ghosts and negatively charged core nanoparticles with small size are preferred as sources of CMNPs due to the well-preserved cell membrane structure of RBC membrane ghosts and their mild binding affinities to small and negatively charged core nanoparticles [68]. TEM or a confocal laser scanning microscope (CLSM) is used to validate the cell membrane integrity in CMNPs; TEM offers a vertical view, and CLSM is diffraction limited and not available for nanoparticle observation [63]. Liu et al. developed a fluorescence quenching assay to observe the full coverage of cell membranes on the surface of core nanoparticles. Fluorescent nitro-2,1,3-benzoxadiazol-4-yl (NBD) was covalently coupled to the core nanoparticles, and integrated cell membrane coating could shield NBD from the reduction by dithionite (DT), which was membrane-impermeable and could quench the fluorescence of NBD-labelled nanoparticles. The NBD fluorescence could remain only in CMNPs with full cell membrane coating in the presence of DT [68]. However, such a fluorescence-quenching assay failed to distinguish partially cell membrane coated-nanoparticles with different surface coverage.

### 3.3. Surface Engineering

Surface engineering of CMNPs plays a crucial role in enhancing their active targeting, improving their pharmacokinetics, and modifying their pharmacodynamics to enhance bioactivity [69]. Surface engineering is generally completed after cell membrane extraction and the fusion of cell membranes with core nanoparticles. In this way, functional moieties can reside at the outer surface of CMNPs. However, the purification operation can be labor-intensive and may alter the physiochemical properties of CMNPs. Another strategy is to directly functionalize the cell membrane of live cells before the cell membrane extraction and the fusion of cell membranes with core nanoparticles [70], which makes the purification operation convenient but imposes uncertainty on the orientation of the engineered cell membrane. Moreover, every step should be cell-friendly during the surface engineering of live cells [71].

Lipid insertion is a non-disruptive approach to equipping CMNPs with functional moieties via lipid anchors [72]. In this strategy, a functional moiety is attached to a lipid molecule, which then guides the functional moiety, such as small molecules or antibodies, to adhere to the cell membrane. However, compared to small molecules, antibodies are larger in size and their geometric orientation is more challenging to control due to the random distribution of functional groups on the protein surface [73]. Lipid insertion can be performed on live cells and cell membrane vesicles [74]. In addition to serving as anchors, the inserted lipids can also function as stimuli-responsive sensors to achieve photothermal conversion [75].

Chemical conjugation is a method of surface functionalization that involves the addition of functional moieties to the surface through strong covalent bonds [76]. Compared to other direct modification methods, which can damage cell membranes and reduce their plasticity, chemical conjugation offers advantages such as high yield synthesis, a wide range of applicability, and easy product separation. Several techniques are commonly employed, including thiol-maleimide coupling, EDC/NHS (1-ethyl-3-(3-dimethyl aminopropyl)-carbodiimide/N-hydroxysuccinimide) coupling, azide-alkyne cycloaddition, and amidation chemistry. Smyth et al. utilized EDC/NHS coupling chemistry to attach alkynyl groups to the surface of extracellular vesicles (EVs), where the amine groups of EV proteins or lipids were cross-linked to 4-pentenyl acids through carbodiimide-activated cross-linking [77]. Microscopy, flow cytometry, and nanoparticle tracking analysis demonstrated that surface functionalization had minimal impact on EV functionality. Chemical conjugation offers the advantage of being easy, quick, and compatible with biomolecules, making it widely utilized for functionalizing biomolecules [78].

Non-covalent adsorption facilitates the binding of functional moieties to CMNPs through hydrogen bonding (dipole-dipole interactions), electrostatic interactions (charge-charge interactions), van der Waals forces, or hydrophobic interactions [70]. Zheng et al. decorated cell membranes with hendeca-arginine peptide (R11) in live bladder cancer cells, and afterward, R11-coated cell membrane vesicles were generated. R11 permeated but did not remain on the cell membrane; therefore, R11 was polymerized with DNA to form nanoparticles so that R11 could be internalized via slow endocytosis rather than fast transduction. Consequently, large amounts of R11 were deposited onto the cell membrane with a right-outside orientation [70].

Genetic modification can present functional moieties on the cell membrane via gene transfection [79]. In the process of genetic modification, robust cell lines can be utilized to permanently express unique surface ligands via viral transfection. These transfected cells can replicate, allowing for maximized population growth, which is beneficial for large-scale manufacturing at a reduced cost compared to other surface engineering techniques [80,81,82,83]. HepG2 cells were genetically modified to express the hepatitis B virus (HBV) preS1 ligand, which was then extracted from the cell membrane to coat oncolytic adenoviruses (OAs). This genetic surface engineering approach has the advantage of decreasing the immunogenicity of OAs and facilitating their targeting of preS1 receptor overexpressed tumors [84]. Genetic modification causes less damage to cell membranes compared to alternative methods, ensuring a more precise surface engineering process.

Metabolic engineering is a technique that manipulates the natural biosynthetic pathways within cells to transport functional moieties onto the cell membrane [85]. More specifically, metabolic substrates are covalently linked with functional moieties, forming conjugates that can be introduced into cells and recycled within the intracellular metabolism [86], and one of their destinations belongs to the cell membrane. Metabolic engineering primarily relies on glycoengineering and lipid engineering. Glycoengineering involves the production of oligosaccharides and glycoconjugates, employing pathways such as the Sialic acid pathway, GalNAc salvage pathway, and Fucose salvage pathway. On the other hand, lipid engineering utilizes natural lipid synthesis pathways, such as the cytidine 5′-diphosphocholine (CDP-choline) pathway [87].

Different methods for cell membrane surface engineering are summarized in Table 2.

## 4. Application of CMNPs in Tackling Urological Diseases

To date, the main types of urological diseases that have been reported to be treated with CMNPs include the following. (1) Cancers: kidney cancer, bladder cancer, and prostate cancer. (2) Inflammations and infections: sepsis and fibrosis. (3) Injury: acute kidney injury, especially ischemia-reperfusion injury. (4) Nephrotoxicity induced by some medicines. (5) Erectile dysfunction.

### 4.1. Cancer Cell Membranes (CCMs)

CCMs have emerged as the ideal candidates for encapsulating nanoparticles in oncological treatment applications for the following reasons. Firstly, CCMs possess remarkable robustness and the ability to proliferate indefinitely [88], making it convenient to obtain and culture their membrane material in vitro. Secondly, CCMs are enriched with a wide array of functional proteins, including membrane proteins that facilitate homologous binding (such as selectins and integrins), biomarkers involved in self-recognition and immune evasion (such as CD47), and tumor antigens associated with immune activation (such as tumor-associated Thomsen–Friedenreich glycoantigens). Notably, when used as a coating, CCMs retain the inherent targeting abilities conferred by a diverse range of cell adhesion molecules, thereby endowing CCM-coated nanoparticles with enhanced cancer-homing features through a homotypic targeting mechanism [89,90]. A study demonstrated that the targeting efficiency of CCMs-coated nanoparticles was 20 times and 40 times higher than that of erythrocyte membrane-camouflaged nanoparticles and bare nanoparticles, respectively [91].

Doxorubicin (DOX) is a common chemotherapeutic drug that can bind and hinder the replication of DNA, thereby inhibiting the cell cycle of cancers. To address its severe side effects, such as impaired bone marrow hematopoietic function, cardiotoxicity, high fever, and so on, which limit its clinical application, Liu et al. cloaked CCM and capped CaCO3 on mesoporous silica NPs (DOX/MSN/CaCO3@CM) [92]. Benefiting from the homotypic targeting abilities of CCMs, DOX/MSN@CaCO3@CM was easily internalized by LNCaP-AI cells (a prostate cancer cell line) and accumulated into the prostate cancer site. The fluorescence intensity of DOX/MSN@CaCO3@CM in the corresponding source cells of LNCaP-AI was much higher than those in the heterotypic cells of MCF-7 (human breast cancer) cells, suggesting the highly specific self-recognition affinity of DOX/MSN@CaCO3@CM to the source cells. CLSM was used to track the DOX red fluorescence in the cells; with the time increased to 8 h, the red fluorescence became dominant in the nuclei rather than in the cytoplasm, which enables DOX to better exert therapeutic effects. A major obstacle in prostate cancer treatment is that most patients eventually develop castration-resistant prostate cancer (CRPC) and have intrinsic or acquired resistance to androgen deprivation therapy (ADT) and other hormone therapy [93]. Therefore, a suitable drug delivery mode is needed for the treatment of CRPC. Lu et al. used CRPC cell membranes as bionic carriers for encapsulating PLGA containing the chemotherapeutic drug docetaxel (DTX) [94]. In addition to properties such as evading early clearance by the immune system and circulatory system and penetrating the extracellular barrier, CRPC cell membranes contain a highly specific library of isotypic molecules that can be used to recognize the same cancer cell types and increase targeted drug delivery by DTX. Measured by flow cytometric analysis to quantify the differences, the cellular uptake of CRPC membrane-coated NPs was approximately 40 times higher than PLGA NPs. Apart from chemotherapy drug delivery, CCMs can also deliver cancer vaccines to fight cancer by activating the body’s immune system. In a recent study [95], Li et al. synthesized PMBEOx-COOH [thioglycolic-acid-grafted poly(2-methyl-2-oxazoline)-block-poly(2-butyl-2-oxazoline-co2-butenyl-2-oxazoline)] to load imiquimod (R837). The surgically harvested CCMs were then coated onto R837-loaded PMBEOx-COOH NPs to obtain surgically derived CCM-coated POxTA NPs (SCNPs/R837). SCNPs/R837 efficiently traveled to the draining lymph nodes and then activated plasmacytoid dendritic cells, triggered a massive release of inflammation-related factors, recruited and activated NK and cytotoxic T lymphocytes cells in prostate cancer lesions, bypassing the tumor immunosuppressive microenvironment and killing tumor cells.

In a study of CMNPs for bladder cancer treatment, a cancer cell membrane-decorated zeolitic-imidazolate framework hybrid nanoparticle (HP) was successfully constructed by Chen et al. [96] co-delivering cisplatin (DDP) and oleanolic acid (OLA). It showed positive results of the platform (HP/DDP/OLA) for the treatment of bladder cancer (BCa) (SW780 cells). HP/DDP/OLA could enhance apoptosis, in detail, after 72 h of incubation at the same drug concentration, the HP/DDP group and HP/OLA group showed 43.6% and 31.4% cell apoptosis, respectively; on the contrary, the HP/DDP/OLA group showed a significant increase to 72.3%. Meanwhile, it can reverse multidrug resistance in SW780 cells more than free drugs alone or mono-delivery systems. Alternatively, to overcome the side effects caused by the permeability barrier and off-targeting of normal urothelial cells and to enhance the efficacy of chemotherapy within BCa. Zheng et al. [70] disguised PLGA nanoparticles containing gemcitabine (PLGA-G) into BCa cell-derived membrane (TM) that were surface-modified with hendeca-arginine peptide (R11). The surface functionalization with R11 endowed TM with the dual-targeting capacity, which originated from the intrinsic BCa targeting capacity of R11 and the homologous tumor targeting capacity of TM. The intravesical drug delivery system comprising R11@TM-camouflaged PLGA-G (R11@TM@PLGA-G) exhibited excellent BCa-targeting capacity and mucus-penetrating efficiency, and even chemo-resected most tumors in murine orthotopic BCa models. It has been acknowledged that the process of endocytosis, specifically through the caveolin-mediated pathway, plays a crucial role in determining the intracellular trafficking of nanoparticles to regions outside of lysosomes. As a result, drugs loaded within CMNPs have the ability to avoid degradation within lysosomes [97].

In conclusion, CCM-coated NPs have been researched for urological cancers targeting to achieve higher on-target payload delivery and lower off-target side effects.

### 4.2. Immune Cell Membranes (ICMs)

Immune cell membrane-coated nanoparticles (ICMCNPs) are emerging, nature-inspired approaches that leverage biocompatibility, prolonged blood circulation time, and enhanced specificity of immune cells to target inflamed tissues and tumors [98]. Each type of immune cell membrane (such as macrophages, dendritic cells, granulocytes, mast cells, and lymphocytes) possesses unique characteristics that can be utilized in membrane-coated nanoparticle-based drug delivery systems [99]. For instance, dendritic cells exhibit specific co-stimulatory/inhibitory molecules on their membranes and play a crucial role in antigen presentation, thus holding promise for DC-based vaccine delivery [100]. T lymphocytes, on the other hand, present T cell receptors (TCRs) on their cell membranes, allowing them to bind specifically and strongly to antigenic determinants found in pathogens, microbes, and tumor cells. Therefore, the membranes of T lymphocytes can be utilized to create efficient nanocarriers for targeted drug delivery [101]. Macrophage derived ICMCNPs can avoid phagocytosis by immune cells via cellular self-recognition mechanisms, and the surface ligands inherited from macrophages can drive ICMCNPs to target diseased sites [102]. Macrophage membrane coated-nanoparticles can respond to multiple abnormal signals, including bacterial toxins, viruses, inflammatory cytokines, cancerous antigens, and so forth [103]. Until now, ICMCNPs that are involved in the treatment of urological diseases have been mainly derived from neutrophils.

Neutrophils are an integral part of the innate immune defense system and are among the early cell types recruited to sites of injury or inflammation [104,105]. This recruitment process involves interactions between P-/E-selectin and their glycosylated ligands, such as P-selectin glycoprotein ligand 1, as well as integrin-mediated adhesion. Leveraging the properties of neutrophils, Liu et al. utilized neutrophil-derived cell membranes to encapsulate Coenzyme Q10 (CoQ10) nanoparticles, creating a formulation known as N-NP CoQ10. This approach aimed to target Ischemia-reperfusion (I/R) injury for therapeutic intervention. Moreover, the functionalization of CoQ10 with neutrophil-derived cell membranes led to synergistic effects in inhibiting oxidative damage and neutralizing pro-inflammatory cytokines at I/R injured sites [106]. In in vitro experiments, the N-NP CoQ10-treated group exhibited 5.32% apoptotic HK-2 cells, while the CoQ10-treated group and NP CoQ10-treated group had 24.9% and 10.3% apoptotic HK-2 cells, respectively. In vivo, N-NP CoQ10 treatment resulted in significantly reduced release of inflammatory cytokines (IL-1β, TNF-α, and IL-6) in I/R injured kidneys, leading to better preservation of kidney function compared to CoQ10 treatment or NP CoQ10 treatment alone.

### 4.3. Stem Cell Membranes (SCMs)

Stem cells possess unique characteristics, including their ability to self-renew and differentiate into specialized cell types [107]. When injured tissues release chemokines, adhesion molecules, and growth factors, stem cells can detect and respond to these signals, migrating toward the sites of injury [108]. Amongst stem cells, mesenchymal stem cells (MSCs) have demonstrated inherent capabilities for homing to tumors and specifically migrating to inflamed tissues. This is facilitated by the interactions between surface receptors on MSCs, such as CXC motif chemokine receptor (CXCR) and CD74, and the corresponding cytokines expressed in cancerous or inflamed tissues [109,110]. Stem cell membrane-coated nanoparticles (SCM-NPs) preserve the surface molecules present on the source stem cell membrane, thereby retaining their intrinsic targeting ability. Additionally, this coating provides protection to the cargo within the nanoparticles, preventing it from being captured by the immune system. By utilizing stem cell membranes as coatings for nanoparticles, SCM-NPs can take advantage of the inherent homing and targeting capabilities of stem cells. This approach offers the potential for targeted drug delivery and therapy, enabling precise delivery to specific tissues or regions of interest.

A novel platform, PDA-Fe3O4@MSC, was developed for the delivery of small interfering RNA (siRNA) across cell membranes [111]. This platform utilized polydopamine-coated iron oxide nanoparticles (PDA-Fe3O4) that were coated with membranes derived from mesenchymal stem cells (MSCs). The incorporated iron oxide nanoparticles served as photothermal agents that could be activated by laser therapy and also acted as magnetic resonance imaging (MRI) trackers. PDA-Fe3O4@MSC combined the therapeutic abilities of gene silencing, photothermal activity, and MRI tracing into a single system for the non-invasive treatment of prostate cancer. In an in vivo antitumor test, the combination of Fe3O4@PDA-siRNA@MSC nanoparticles with laser irradiation resulted in a 60% reduction in tumor volume after a 15-day therapy period. Another multifunctional platform (PDA-DOX/siRNA@SCM NPs) is introduced by Mu et al. to combine chemotherapy and gene therapy for prostate cancer treatment. Considering the adaptive upregulation of programmed cell death ligand 1 (PD-L1) during Dox-based chemotherapy, this platform co-delivered PD-L1 siRNA and Dox into prostate cancer cells. The goal was to eliminate the expression of the PD-L1 protein, thereby restoring the immune anti-tumor activity of T cells while maximizing the anti-cancer effects of Dox. The SCM coating on the nanoparticles reduced the clearance of Dox from the bloodstream, resulting in higher Dox accumulation in tumors compared to the Dox group and PDA-DOX group. Additionally, the SCM coating alleviated the acute toxicity of Dox without causing a significant decrease in body weight over time [112].

The value of SCM-coated nanoparticles for drug delivery is discounted by the relatively shorter circulation time of SCM-coated nanoparticles, compared with red blood cell membrane-coated nanoparticles. In addition, it is inconvenient to obtain large numbers of stem cells for large-scale production of SCM-coated nanoparticles [113].

### 4.4. Red Blood Cell Membranes (RBCMs)

Red blood cells (RBCs) have an average lifespan of about 120 days, making them an ideal choice for the preparation of long-circulating nanoparticles [114]. RBCs were the first type of cells used to create membrane biomimetic carriers, as demonstrated by Hu et al. in 2011 [47]. In their study, encapsulating nanoparticles with RBC membranes (RBCMs) extended the blood circulation time from 24 h to 72 h. RBCM-coated nanoparticles retain the natural surface structure of RBCs and preserve the presence of CD47, a protein that interacts with the SIRP-α receptor to inhibit phagocytosis by macrophages. This interaction releases a “don’t eat me” signal, contributing to the nanoparticles’ ability to evade macrophage uptake [69,115]. It has been observed that the expression level of CD47 is down-regulated on aging RBC membranes, which ultimately leads to their phagocytosis by macrophages and accumulation in the liver. Leveraging the natural properties of aging erythrocytes, aging RBCMs can serve as carriers for drug delivery targeting detoxification and liver-specific therapeutic purposes [116].

Although there have been limited advances in the study of RBCM-coated nanoparticles for treating urological diseases, they hold the potential for mitigating or even eliminating drug-induced nephrotoxicity. Antibiotics are the first choice for the treatment of sepsis and septicemia, but high doses of antibiotics can lead to serious adverse toxic reactions and side effects [117]. To address this issue, Liu et al. synthesized RBCM-coated nanoparticles (RBCNPs) modified with γ3 peptide (γ3-RBCNPs) for targeted therapy of *Klebsiella pneumonia*-induced sepsis. The γ3 peptide specifically binds to ICAM-1 at the infection sites, allowing γ3-RBCNPs to transport ciprofloxacin to the infection regions rather than accumulating in the kidneys. In another study, Su et al. introduced RBCM-coated gelatin nanoparticles loaded with berberine hydrochloride (RBGPs) to achieve sustained release of the drug and ensure biosafety [118]. Human embryonic kidney (HEK 293T), a standard cell model for the nephrotoxicity evaluation in vivo, was incubated with different concentrations of RBGP (30, 60, 120, 240, 300, 480, 600, and 750 μg/mL) for 24 h. The RBGPs with the highest concentration (750 μg/mL) did not hamper the cell viability of HEK 293T cells, while free BH will cause cytotoxic effects on liver and kidney cells at 50 ug/mL [119]. This indicated that RBGPs improved the biocompatibility of their loading drugs to reduce nephrotoxicity. Similarly, Malhotra et al. derived amphiphilic fluorophore-labeled nanovesicles (NVEs) from RBCMs [120] to load the hydrophobic drug camptothecin (CPT). Compared with free CPT, NVEs-coated CPT showed higher retention in the circulation over 48 h and insignificant accumulation in kidneys.

Due to the deformability of RBCNPs, they are excellent fits for the delivery drugs across physiological barriers. However, dense tumor stroma contributes to the impermeability of cancerous tissues [30], thus blocking the penetration of drugs, nanoparticles, or even highly deformable RBCNPs. Zhou et al. anchored recombinant human hyaluronidase, PH20 (rHuPH20) onto the surface of RBCNPs to degrade hyaluronic acid (HA) [121,122], which is highly expressed in the extracellular matrix (ECM) of approximately 80% of prostate cancer [122]. The diffusion co-efficiency of rHuPH20-Anchored RBCNPs nearly doubled over that of RBCNPs, and they were preferably internalized by PC3 prostate cancer cells.

### 4.5. Extracellular Vesicles (EVs)

EVs are structures surrounded by a lipid bilayer and are released by both prokaryotic and eukaryotic cells [123,124]. According to guideline of International Society for Extracellular Vesicles, EVs is a collective term covering various subtypes of membranous structures released by cells, including exosomes, microvesicles, microparticles, ectoderm, epithelium, apoptotic bodies, and many others [125]. Currently, the most relevant to the treatment of urological diseases are exosomes (Exos), microvesicles (MVs) [126]. Exosomal formation is regulated by endosomal sorting complexes required for transport (ESCRT) proteins; therefore, these proteins and their accessory proteins (Alix, TSG101, HSC70, and HSP90β) can be found in exosomes regardless of the type of cell from which they originate. Thus, this set of proteins is often regarded as “exosomal markers”. Exosomes not only participate in cell–cell communication, tumor progression, and stimulation of immune responses by acting as antigen-presenting vesicles [127], but also act as carriers of biomarkers for diseases [128,129,130], such as Parkinson, glioblastoma, acute kidney injury, etc. The marker proteins associated with MVs mainly include cytosolic proteins and proteins associated with the plasma membrane [131], such as tetraspanins, cytoskeletal proteins, heat shock proteins, integrins, glycan-binding proteins, etc. MVs also play a crucial role in cell–cell communication, facilitating interactions between local cells as well as between local and distant cells. They have the ability to package various cargoes such as proteins, nucleic acids, and lipids and transport them to recipient cells [132]. In contrast to exosomes and MVs, apoptotic bodies contain intact organelles, chromatin, and small amounts of glycosylated proteins. Apoptotic bodies are involved in important biological processes, including the clearance of apoptotic cells and intercellular communication [133]. It is worth noting that the biggest difference between EVs and other membrane vesicles is that EVs accommodate ample intracellular components with multiple biological functions [134], including lipids, proteins, sugars, and RNAs. Unique composition (more enriched lipids compared to the plasma membrane, e.g., cholesterol, phosphatidylserine, glycosphingolipids, sphingomyelin, and unsaturated lipids) of EVs membrane endows EVs with high deformability to cross physiological barriers [135] with stability both in circulation and in vitro [136,137].

The fusion process of nanoparticles to EVs mainly includes the passive loading of nanoparticles into EVs via sonication, repeated mechanical extrusion and electroporation, and active excretion of nanoparticles into EVs after the endocytosis of nanoparticles into parental cells [138,139,140]. Huang et al. reported an effective transportation system utilizing both mesenchymal stem cells and their secreted MVs to contain gold nanostars (GNS) and intracellularly assemble GNS into clusters for targeted photothermal therapy of prostate cancer. Under the irradiation of a near-infrared ray, MSCs mobilized GNS-loaded EVs excretion to infiltrate tumors. In vivo intratumoral distribution assessment revealed that localized GNS-generated signal spots within free GNS-treated tumors covered an area of 0.022 cm^2^, while those within the transportation system-treated tumors were uniformly distributed throughout the tumor with an area of 0.073 cm^2^. The anti-cancer effect of the transportation system was the strongest among all types of therapy applied in this study including GNS-treated group and PBS control group [141]. Saari et al. demonstrated [142] that EVs are capable of delivering loaded drugs to parental cells via an endocytic pathway. While empty EVs increased tumor cell viability, EVs loaded with paclitaxel still exerted enhanced cytotoxicity. Pan et al. utilized urine-derived exosomes to encapsulate nano-sized Fe_3_O_4_ integrated with DOX (Exo/Fe3O4@Dox) as a chemo/chemodynamic therapeutic nanoplatform for targeted treatment of prostate cancer. In 3D cell spheroid assays, Exo/Fe_3_O_4_@Dox exhibited deeper penetration compared to Fe_3_O_4_@Dox and free Dox [140]. Zhou et al. created EVs from macrophages that co-deliver CD73 inhibitor (AB680) and antibodies targeting PD-L1 (aPDL1) [143]. AB680 inhibits extracellular cytidine production and allows activation of cytotoxic T lymphocytes, providing synergistic efficacy with aPDLl in the fight against bladder tumors. Extracellular vesicles from urine stem cells (USC-EVs) deliver Hyaluronic acid (HA) and have been shown to be effective in erectile dysfunction (ED) in type 2 diabetic treatment [144]. Compared to the HA group, ED was improved in the USC-EVs-HA group by promoting the proliferation of endothelial cells and smooth muscle in the corpus cavernosum.

The dose-dependent nature of glucocorticoids can produce serious side effects (osteoporosis, concentric obesity, infection, etc.) in clinical applications of treatment for renal inflammation [145,146]; therefore, Tang et al. used macrophage-derived MVs delivered with dexamethasone (DEX) to treat renal inflammation and demonstrated superior ability to suppress renal inflammation compared to DEX treatment alone, without obvious adverse effects. LFA-1 and VLA-4 present on MV-DEX were responsible for their homing to the inflamed kidney. In addition, adjunct proteins in the MV-DEX could compensate for the loss of the receptor in kidneys and improve susceptibility to glucocorticoids, which would benefit many steroid-resistant patients [147]. Moreover, glucocorticoid receptors encapsulated in EVs were also delivered to the receptor cells, thereby enhancing cellular sensitivity to dexamethasone treatment [147]. In addition to glucocorticoids, RNA interference therapeutics, such as small interfering RNA (siRNA) and microRNA (miRNA), offer a more specific and potent approach to modulating inflammation-associated genes. EVs have been harnessed as delivery vehicles for siRNAs to injured tubules, resulting in the dual suppression of transcription factors and attenuating renal inflammation, fibrosis, and the transition from acute kidney injury (AKI) to chronic kidney disease (CKD) [148]. RVG peptide-modified EVs loaded with miR-29 demonstrated tropism for the injured kidney in mice and improved renal fibrosis by downregulating YY1 and TGF-β3 [149]. Diao et al. developed a novel strategy by using the polycationic membrane-penetrating peptide TAT to encapsulate siRNAs into EVs. Simultaneous knockdown of FLOH1, NKX3, and DHRS7 genes using siRNA showed potential for improving treatment in CRPC [150]. Kurniawati et al. utilized MSC-derived exosomes as exogenous vectors to deliver microRNA-let-7c, attenuating CRPC aggressiveness and significantly reducing cell proliferation and migration in CRPC cells [151]. Zhupanyn et al. combined polyethyleneimine (PEI) nanoparticles with EVs, harnessing the beneficial properties of both systems for the delivery of siRNAs and antimiRs, resulting in a significant inhibition in loaded PC3 cells [152]. EVs enhance the performance of PEI nanoparticles and show significant inhibition in loaded PC3 cells. In addition to synthetic therapeutics, natural anti-inflammatory materials present in the body can be exploited for disease therapy [153]. For instance, inhibitors of NF-κB (IκB) proteins, when engineered into a nondegradable super-repressor form (srIκB), can sequester NF-κB in the cytoplasm. Using EVs as carriers, srIκB was efficiently packaged and delivered to neutrophils and macrophages, ameliorating inflammation by inhibiting NF-κB signaling in sepsis and ischemia-injured kidneys [154].

Besides acting as nanoparticles and therapeutic materials delivery vehicles, EVs derived from certain cell types (e.g., stem cells, immune cells, tubular cells, cancer cells) possess intrinsic self-therapeutic efficacy and can be utilized directly as therapeutic biomolecules [155,156,157,158,159]. Through the presence of natural surface proteins (e.g., integrins, L-selectin, CD44, CXCR4) or engineered targeting moieties (e.g., Kim-1-binding peptide, RVG peptide), EVs exhibit efficient localization to the diseased kidney, leading to improvements in renal function and injury by inhibiting apoptosis, inflammation, and fibrosis, while promoting cell proliferation, angiogenesis, and autophagy. For example, EVs derived from mesenchymal stem cells exert protective immunomodulatory effects through the release of IL-10, a crucial anti-inflammatory mediator, both in vitro and in mouse models of bacterial-induced sepsis [160], so that sepsis-associated acute kidney injury (SA-AKI) can be alleviated and treated.

However, there are still many unresolved issues, for example, cancer cell-derived EVs have been suggested to promote cancer survival and proliferation [161], and EVs may amplify urological damage and contribute to the progression of urological diseases due to their roles in cell-to-cell and organ-to-organ crosstalk. In addition, distinguishing from other cell membranes, EVs actually cannot evade the own immune system effectively [162]. Meanwhile, there is a lack of sufficient preclinical experiments in the application of disease treatment, and manufacturing and engineering are indeed complex and challenging. However, they are by no means insurmountable, and we can expect rapid expansion in the realm of EV-based therapy in urological diseases.

### 4.6. Other Membranes

Bacterial outer membrane vesicles (OMVs) have emerged as a novel approach in the design of CMNPs. Both Gram-negative and Gram-positive bacteria can release OMVs, which are enriched with bioactive proteins, toxins, virulence factors, and immunogenic substances that play important roles in bacterial-host interactions [163]. The non-replicative nature of OMVs contributes to their generally safe use in vivo [164]. OMVs possess selective permeability and enable cellular communication, making them advantageous as nanocarriers in biomedical applications. The OMV-coated nanoparticles exhibit excellent drug-loading capacity and selective permeability to bacterial cell membranes [165] because bacteria do not attack or block OMVs owing to their self-recognition functions and intercellular communication functions [166]. Taking advantage of this, Gao et al. developed an active targeting delivery system by coating antibiotic-preloaded nanoparticles (NP-Antibiotic) with the membrane of EVs secreted by *Staphylococcus aureus*. The resulting NP-Antibiotic@EV particles mimicked *S. aureus* and actively targeted *S. aureus*-infected macrophages in vitro. These antibiotic-loaded CMNPs exhibited potent intracellular elimination of *S. aureus*, comparable to or even better than the effects of their individual antibiotic cargoes, despite the slow release of the antibiotics [167]. Importantly, NP@EV showed improved efficacy in mitigating metastatic foci of infection in major organs, particularly in the kidney, which is the organ with the highest bacterial burden and the highest risk of *Staphylococcus aureus* infection.

Platelets (PLTs) are small cytoplasmic fragments that are released from mature macrophages in the bone marrow of mammals. One unique feature of platelets is the expression of CD47, which allows platelet-derived cell membrane-coated nanoparticles to evade uptake by macrophages. The surface of platelet-derived cell membranes is enriched with various proteins, including P-selectin, CD40, CD55, and CD59, which play important roles in modulating disease progression and inhibiting the activation of the immune complement system [54,168]. P-selectin, in particular, can bind to CD44, which is highly expressed on the membranes of tumor cells. This property enables platelet-derived cell membrane-coated nanoparticles to target tumors effectively. Thanks to the interactions between specific receptors and glycoproteins that mediate their strong adhesion to damaged arteries [53], PLTs-coated NPs can target atherosclerosis and bacterial infections.

Examples of CMNPs as drug carriers for the treatment of urological diseases are summarized in Table 3.

## 5. Prospects and Challenges

CMNPs, this biomimetic structure preserves the surface properties and functions of the source cells. Cell membranes with biological properties can well compensate for the instability of nanoparticles and the disadvantages of adverse effects on the organism, as mentioned in this review, which can prolong the circulation time of nanoparticles, evade clearance by the immune system and target the drug transport and release to the corresponding organ tissues and lesions. Benefiting from these properties, it provides an efficient and biocompatible drug delivery strategy for the treatment of various urinary system diseases such as urinary system tumors, inflammation, and acute kidney injury.

After our study, we found that single-cell membrane coating still has limitations for enhancing the utilization of nanoparticles (Table 4). For instance, RBCM-coated nanoparticles prolong blood circulation, but they lack targeting capabilities [169]. As mentioned earlier, surface engineering of CMNPs can enhance their active targeting, pharmacokinetics, and biological activity. Hybrid cell membrane coating represents a promising approach [170], as it ideally integrates diverse biological functions. In 2018, Zhang and colleagues reported an innovative bionanotechnology for membrane hybridization [171]. They developed an erythrocyte-platelet hybrid membrane-camouflaged nanoparticle, incorporating surface membrane protein markers derived from both cell types. When compared to single-membrane-coated formulations using only erythrocytes or platelets, the resulting dual membrane-coated nanoparticles exhibited remarkable long circulation and distribution in mouse models. Subsequently, hybrid membranes combining erythrocytes with cancer cells, platelets with neutrophils, and cancer cells with platelets were successfully fabricated for applications in fields such as individualized cancer therapy [172,173]. The potential for exploring countless different combinations is vast, and this may eventually lead to the development of novel multifilm nanoparticle platforms.

Although much has been achieved with CMNPs, there remain some challenges to be overcome. First and foremost, the primary concern when utilizing cell membranes, such as CCMs or EVs, is ensuring biosafety. These membranes themselves possess the potential to facilitate tumor growth or disease progression. Inappropriate development of immunogenicity can induce a harmful immune response when using bacterial membranes [161]. Additionally, the application of RBCMs can lead to hemolysis during transfusion if there is a blood group mismatch, activating the host immune system [174]. Even in the case of normal cell membranes, it is essential to consider their long-term safety during clinical implementation, as there may exist a biological disparity between the native cell membrane and the extracted formulation.

Furthermore, in relation to operational techniques, additional research and improvements are required regarding the integrity and orientation of cell membrane coatings [62,68]. This is crucial since the functionality of cell membranes is largely determined by their intactness and proper orientation. Additionally, functional surface proteins are frequently inactivated when treated with lysis buffer and hypotonic solution under in vitro conditions [175]. Challenges may also arise in terms of sourcing and culturing cell membranes for larger-scale production. For instance, certain cell membranes such as SCMs may not be easily accessible, and the cultivation of red blood cells and platelets necessitates a blood supply.

However, despite the promising results obtained in animal experiments, particularly in mice, there remain numerous areas that require further investigation and exploration before the technology can be effectively applied in clinical settings for the benefit of patients. Currently, CMNPs have not been utilized in clinical practice, indicating that the field is still in its early stages and has yet to mature to the level of translational research, encompassing the transition from laboratory experiments to clinical trials. To facilitate their clinical translation, it is crucial to establish scalable and easily reproducible manufacturing practices that can expedite the process.

## 6. Conclusions

Overall, naturally derived cell membranes possess inherent functional properties that enhance the biopharmaceutical performance of encapsulated nanoparticles. This makes Cell Membrane Coated-Nanoparticles (CMNPs) a promising avenue for various applications, including drug delivery, disease prevention, and treatment. The potential of CMNPs in the medical field is extensive, and despite some existing limitations, their remarkable advantages pave the way for targeted treatments of urological diseases. In the future, further innovative strategies will be explored to unlock new possibilities for CMNPs in the treatment of kidney and urological diseases.

## Figures and Tables

**Table 1 pharmaceutics-15-01899-t001:** Comparison of different fusion methods of CMNPs.

Fusion Methods	Process	Advantages	Limitations	References
Extrusion	NPs and the extracted purified cell membranes are co-extruded through a porous membrane using extruders	Simple and practical operation steps	Inefficient synthesis Low rate of synthesis	[46,51]
Ultrasound	The mixture formed by mixing cell membranes and NPs is sonicated at a certain frequency for a certain period of time	Reducing loss of raw materials for mass production	Affecting the size and stability of core NPs Potentially disrupting NPs	[52]
Microfluidic electroporation	Mix cell membranes and NPs in a microfluidic chip, then flow through the electroporation zone, inducing the formation of micropores on the cell membrane	Avoiding the destruction of NPs Less time-consuming and labor-intensive	Complex operating procedures	[53]
In situ polymerization	Using cell membranes to template inner core NPs, polymerize NPs by the action of initiators	Ensuring the integrity of the coating Easy control of the size and stiffness of CMNPs with no easy leakage of contents	Small application range High selectivity for core NPs	[57]

**Table 2 pharmaceutics-15-01899-t002:** Methods for cell membrane surface engineering.

Method	Mechanism	Superiority	Deficiency	References
Lipid insertion	Ligands or therapeutic molecules are anchored to the cell membrane via lipid–lipid interactions	Maintaining the integrity of content while avoiding complex steps	Not suitable for large transmembrane protein receptors or ligands	[71]
Chemical conjugation	Adding functional moieties by strong covalent connection on the surface	High yield, wide in scope, and easy product separation	Affecting protein integrity and function potentially by the use of chemical reagents	[75]
Non-covalent adsorption	Non-covalent and weaker binding of functional moiety to CMNPs	Enhancing the surface function without affecting the orientation of cell membrane coating	Poor adhesion and unable to maintain long-term stability	[69]
Genetic modification	Presenting functional moieties on the cell membrane via gene transfection	Functionalizing cell membranes precisely in a non-invasive strategy	Not suitable for small molecule therapeutic factors or ligands and operation is challenging	[78]
Metabolic engineering	Manipulating cellular natural biosynthetic pathways to transport functional moieties onto the cell membrane	Allowing for straightforward cell membrane functionalization by endogenous processes in cells	Difficult to control splice site specificity and efficiency	[84]

**Table 3 pharmaceutics-15-01899-t003:** Applications of diverse sources of cell membrane in urological diseases.

Sources of Membrane	Cargoes/Nanoparticle	Diseases/Effect	Properties	References
LNCaP-AI cell	DOX/MSN	PCa	Adhesion to targeted tumor sites Proteins that mediate homologous binding Promotes tumor-specific immunity	[91]
DU145 cell	DTX/PLGA	CRPC		[93]
Surgically derived cancer cell	Imiquimod/PMBEOx-COOH	Pca		[94]
BCa cell	Cisplatin and oleanolic acid/Hybrid nanoparticle	BCa		[95]
BCa cell	Gemcitabine/PLGA	BCa		[69]
Neutrophil	Coenzyme Q10/PEG-PLA	Renal ischemia-reperfusion injury	Good biocompatibility Mitigates the inflammatory conditions and tumors specifically Produces toxic molecules to quickly eradicate the phagocytosed pathogen	[105]
RBC	BH/Gelatin	Achieve sustained release and reduce the nephrotoxicity of BH	Prolonged blood circulation CD47 expression Immune evasion	[117]
RBC	Ciprofloxacin/PLGA	Klebsiella pneumoniae-Induced sepsis		[116]
RBC nanovesicles	Camptothecin/-	Reduce the accumulation of camptothecin in the kidneys		[119]
RBC	-/PLGA	PCa		[121]
Mesenchymal stem cell	siRNA/Fe3O4@PDA	PCa	Penetrates across the endothelium May target particular tumor Homing ability	[110]
Stem cell	DOX and PD-L1 siRNA/Polydopamine	PCa bone metastases		[111]
MSC-Derived MVs	-/Gold nanostars	Photothermal Therapy of PCa	Possess functional intracellular components Inheritance of parent cell characteristics High deformability to cross physiological barriers Therapeutic biomolecules directly	[140]
Cancer cell-derived MVs and EXOs	Paclitaxel/-	PCa		[141]
Urinary exosomes	DOX/Fe3O4	PCa		[139]
MSC-derived Nanovesicles	IL-10/-	Alleviate and treat sepsis-associated acute kidney injury		[159]
Macrophage-Derived exosome-mimetic nanovesicles	Monoclonal antibody to PD-L1 and CD73 inhibitor	Immunotherapy strategy for bladder cancer		[142]
Macrophage-derived MVs	Dexamethasone/-	Renal inflammation and fibrosis		[146]
RBC-derived-EVs	Transcription factors P65 and snai1 siRNA	Acute kidney injury		[145]
EXOs	miRNA-29	Kidney Fibrosis		[147]
EVs	siRNA	CRPC		[148]
MSC-exosome	miRNA-lethal 7c	CRPC		[150]
EVs	siRNA/Polyethylenimines	PCa		[151]
EXOs	inhibitor of NF-κB	Sepsis and ischemia-injured kidneys		[153]
Urine-derived stem cells EVs	HA	Erectile dysfunction		[143]
Bacterium	Gold nanoparticles	Mitigate metastatic foci of infection in kidneys	Immune stimulation	[166]

Abbreviations: Prostate cancer (PCa); Bladder cancer (BCa); Red blood cell (RBC); Doxorubicin (DOX); Mesoporous silica nanoparticles (MSN); Docetaxel (DTX); Castration-Resistant Prostate Cancer (CRPC); Polylactic-glycolic acid (PLGA); poly(2-methyl-2-oxazoline)-block-poly(2-butyl-2-oxazoline-co-2-butenyl-2-oxazoline(PMBEOx-COOH); Berberine Hydrochloride (BH); polydopamine (PDA)-coated hydrophobic Fe_3_O_4_ NPs (Fe_3_O_4_@PDA); Small interfering RNA (siRNA); Micro RNA (miRNA); Programmed cell death ligand 1 (PD-L1); Extracellular vesicles (EVs); Microvesicles (MVs); Exosomes (EXOs).

**Table 4 pharmaceutics-15-01899-t004:** Summary of different features of cell membranes in CMNPs.

Cell Membranes	Typical Biomarkers	Advantages	Limitations	References
CCMs	Selectins, Integrins, CD47, and TAG	Homologous targeting and culture conveniently in vitro	Potential causes of tumor metastasis	[87,88,89,90]
ICMs	CD45, CD47, TCRs, Co-stimulatory/inhibitory molecules	Immune escape and good biocompatibility	Complexity of extraction, immunogenicity	[97,98,99,100,101,102]
SCMs	CD74, CXCR, and Other chemokine	Particular tumor homing ability and inflammatory migratory	High cost of preparation and low specificity	[106,107,108,109,112]
RBCMs	CD47	Long circulation time and simple for surface engineering	Lack of targeting capabilities	[46,68,114]
EVs	ESCRT protein and Accessory proteins	High deformability and inheritance of parent cell	Lack of immune evasion and may promote disease progression	[125,126,127,128,129,130,131,132,133,134,135,136]
OMVs	Virulence actors	Immune activation	Insecurity in vivo	[162,163,164,165]
Platelets	P-selectin, CD47, CD55 and CD59	Tumor and inflammation targeting	Instability	[52,53,167]

## Data Availability

Data sharing not applicable—no new data generated.

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
