# Peer review of "Recent Advances in Cell Membrane Coated-Nanoparticles as Drug Delivery Systems for Tackling Urological Diseases"

_pharmaceutics, 2023, doi:10.3390/pharmaceutics15071899_

Round 1

Reviewer 1 Report

The authors discussed the fabrication of cell membrane-coated nanoparticles as DDS and their application for treatment of urological diseases. It is informative and well written. I recommend its publication after addressing the following concerns.

1. It is better to summarize the types of urological diseases  that have been reported to be treated by cell membrane-coated nanoparticles.

2. It is better to explain why it is necessary to coat nanoparticles by cell membrane in the introduction, such as fast removal of nanoparticle by RES (Cancer Res (1982) 42 (4): 1412–1422.).

3. Please summarize the targeting mechanism for different urological diseases and discuss the multifunctionality of  cell membrane-coated nanoparticles (https://doi.org/10.1021/jacs.0c09029).

4. Extracellular vesicles actually can not evade the own immune system (https://doi.org/10.1016/j.jconrel.2016.01.009). It deserves to be mentioned.

5. In fact, cell-membrane coated nanoparticles always fail to give the stealth effect. It is still very challenging to employ cell membrane coating technology. Please refer a recent paper that comprehensively analyzed cell-membrane coated nanoparticles (10.1016/j.addr.2023.114895) and provide a deep discussion based on your own thoughts. 

Author Response

Response to Reviewer 1 Comments

Point 1: It is better to summarize the types of urological diseases that have been reported to be treated by cell membrane-coated nanoparticles.

Response 1: Thank you for your kind suggestion. The types of urological diseases that have been reported to be treated by CMNPs mainly include: 1) Cancers: kidney cancer, bladder cancer and prostate cancer. 2) Inflammations and infections: sepsis, fibrosis. 3) Injury: acute kidney injury especially ischemia-reperfusion injury. 4) Nephrotoxicity induced by some medicines. 5) Erectile dysfunction. We have summarized it in the section “4. Application of CMNPs in Tackling Urological Diseases” in line 286-290. Thank you!

Point 2: It is better to explain why it is necessary to coat nanoparticles by cell membrane in the introduction, such as fast removal of nanoparticle by RES (Cancer Res (1982) 42 (4): 1412–1422.).

Response 2: Thank you for you suggestion. We have explained why it is necessary to coat nanoparticles by cell membrane in the introduction in line 28-30. “Nanoparticles will be recognized and fast removed by the reticuloendothelial system (RES) as foreign elements in the body, causing immune responses and toxic effects, which will be the major hurdles that almost all platforms must overcome.” Thank you so much.

Point 3: Please summarize the targeting mechanism for different urological diseases and discuss the multifunctionality of  cell membrane-coated nanoparticles (https://doi.org/10.1021/jacs.0c09029 ). PMID: 33370092 DOI: 10.1021/jacs.0c09029

Response 3: Thank you for your comment. The targeting mechanism for urological diseases are summarized in line 88-92 as follows:

  1. Nanomaterials rely on enhanced permeability and retention (EPR) effect to passively target urological lesions especially tumors. (PMID: 31794153 PMCID: PMC7224408 DOI: 10.1002/adhm.201901223)
  2. Specific receptors, such as fibroblast growth factor receptor, present on the surface of tumor cells can achieve targeted delivery of nanomedicines by binding to functionalized ligands. (PMID: 32767764 DOI: 10.3322/caac.21631)
  3. Tissue microenvironment (PMID: 33370092 DOI: 10.1021/jacs.0c09029): Diseased tissues have a distinct pathophysiological microenvironment owing to metabolic reprogramming, including redox environment, acidity, hypoxia, and protein (e.g., protease, growth factor, and receptor) expression/secretion.
  4. Renal or other urological inflammatory diseases release inflammatory factors to aid in targeted drug delivery.

CMNPs exhibit a biomimetic structure that retains the surface characteristics of the source cells and imparts bioactivities to the encapsulated nanoparticles, including immune evasion, cell-specific targeting, and extension of systemic circulation ( Line 94-96). We have added these precious suggestions and comments in the article. Thank you so much!

Point 4: Extracellular vesicles actually can not evade the own immune system (https://doi.org/10.1016/j.jconrel.2016.01.009). It deserves to be mentioned.

Response 4: Thank you for your valuable comment. After our search and reflection, we have mentioned the critical point “EVs actually can’t evade the own immune system effectively” in the part of “4.5 Extracellular vesicles (EVs)” in line 586-587. Thank you so much!

Point 5: In fact, cell-membrane coated nanoparticles always fail to give the stealth effect. It is still very challenging to employ cell membrane coating technology. Please refer a recent paper that comprehensively analyzed cell-membrane coated nanoparticles (10.1016/j.addr.2023.114895) and provide a deep discussion based on your own thoughts. 

Response 5: Thank you for your comment. According to recent research findings, more than 85% of the reported stealth na-nomaterials encounter a rapid drop of blood concentration to half of the administered dose within 1 h post administration which means that nanoparticles did not achieve a good stealth effect. We concluded that PEGylation is a commonly used method to ex-tend blood circulation, but membrane coting is more in line with the concept of holism because of the fine structural hierarchy of biological systems. We have added it in the article in line 31-37. Thank you!

Thanks very much for taking your time to review our manuscript (pharmaceutics-2447937) again. Your comments and suggestions are all valuable and meaningful for improving our paper! We have studied all reply carefully and have made conscientious correction.

Kind regards

Pu Zhang, Zhejiang Provincial People's Hospital

Reviewer 2 Report

Cenchao Yao, Dahong Zhang, Heng Wang, and Pu Zhang:

Recent Advances in Cell Membrane Coated Nanoparticles as Drug Delivery Systems for Tackling Urological Diseases

The matter of the special field is well summarized. I have found very positive, that the authors compared the different coating materials critically and give information about the problems which must be searched. Beside several advantages of the manuscript, I suggest inserting of some new points and further information whereby the scientific sound of the work would be increased.

Remarks suggestions are the follows.

The enhanced permeability and retention (EPR) effect is widely used idea for passive targeting. I suggest the insertion of the information for this size limitation. Is this characteristic size-range general, or it is different for urological diseases? Do the coated nanoparticles exhibit the same size range (may be further data in Table3). Are there special criteria for CMC nanoparticles used urological diseases comparing with other diseases? In the literature the descriptions about the location of CMC nanoparticles - uptakes, are rather incomplete. Can the information about the (molecular) specific targeting of intracellular compartments (especially in cancer cells) be more highlighted?

Is the description in Table 1. correct (Coring NPs)? I suggest the presentation of the main features (typical components, ratios) of the described membrane types (cancer cell membranes, immune cell membranes, stem cell membranes, red blood cell membranes, membrane of extracellular vesicles), in a form of a Table.  I note that the International Society for Extracellular Vesicles offered a new differentiation for extracellular vesicles.  

 I am not qualified to assess the quality of English in this paper , but the manuscript reads

Author Response

Response to Reviewer 2 Comments

Point 1: The enhanced permeability and retention (EPR) effect is widely used idea for passive targeting. I suggest the insertion of the information for this size limitation. Is this characteristic size-range general, or it is different for urological diseases? Do the coated nanoparticles exhibit the same size range (may be further data in Table3). Are there special criteria for CMC nanoparticles used urological diseases comparing with other diseases?

In the literature the descriptions about the location of CMC nanoparticles - uptakes, are rather incomplete. Can the information about the (molecular) specific targeting of intracellular compartments (especially in cancer cells) be more highlighted?

Response 1: Thank you for your so kind comments and corrections.

  1. NPs with a hydrodynamic diameter (HD) of 100–400 nm have previously been considered optimal for passive tumor targeting due to the enhanced permeability and retention (EPR) effect; however, Cabral et. al has recently reported that 30 nm micelles could show much more effective tumor penetration than 100 nm micelles (PMID: 22020122). Even small sized PEGs with MW of 5 kDa exhibited reasonable uptake in the tumor site via the EPR effect, and 20 kDa PEG with 11 nm in hydrodynamic diameter (HD) showed the best performance and maximized TBR with rapid renal excretion. ( PMID: 7640051 DOI: 10.1016/0959-8049(94)00514-6)(line 74-79)
  2. The size of CMNPs is similar to or within an acceptable range of original NPs, for example (DOI: 10.3390/pharmaceutics11020093), the hydrodynamic diameters of BGPs increased from 243.6 ± 3.7 nm to 260.3 ± 4.1 nm upon coating with RBCM. But only some researches mentioned in the section “4. Application of CMNPs in Tackling Urological Diseases” has been analyzed for the characterization of CMNPs.
  3. The CMC nanoparticles used urological diseases don’t have special criteria comparing, but Seymour et al.demonstrated that large copolymers (MW >40 kDa) are unable to be excreted via the kidney filter and urinary track, and thus persist in circulation, whereas smaller copolymers (MW <40 kDa) are subject to rapid renal clearance.( PMID: 7640051 DOI: 10.1016/0959-8049(94)00514-6)
  4. For nanoparticles, nanomaterials correct the intracellular trafficking of drugs into the expected subcel-lular compartments. (doi:10.1039/c5nr05139h\doi:10.1016/j.cell.2015.05.025\doi:10.1039/c8bm00381e.)

As for CMC nanoparticles, it has been recognized that endocytosis through the caveolin-mediated pathway determines the intracellular trafficking of nanoparticles to nonlysosome-localized regions(PMID: 33859387 DOI: 10.1038/s41551-021-00701-4). Therefore, the drugs loaded in CMNPs were capable of escaping the fate of degradation in lysosomes.

In another article(PMID: 30572156 DOI: 10.1016/j.colsurfb.2018.12.038), CLSM was used to track the DOX red fluorescence in the cells, with the time increased to 8 h, the red fluorescence became dominant in the nuclei rather than in the cytoplasm, which enables DOX to better exert therapeutic effects (316-319)

We have enriched the article according to you the above valuable suggestions. Thank you so much.

Point 2: Is the description in Table 1. correct (Coring NPs)? I suggest the presentation of the main features (typical components, ratios) of the described membrane types (cancer cell membranes, immune cell membranes, stem cell membranes, red blood cell membranes, membrane of extracellular vesicles), in a form of a Table.  I note that the International Society for Extracellular Vesicles offered a new differentiation for extracellular vesicles.  

Response 2: Thank you for your kind comments and corrections. We have modified some of the contents of Table 1 and replaced Coring NPs with NPs. We have summarized the different features of cell membranes in CMNPs in Table 4. And according to guideline of International Society for Extracellular Vesicles, extracellular vesicles (EVs) is a collective term covering various subtypes of membranous structures released by cells, including exosomes, microvesicles, microparticles, ectoderm, epithelium, apoptotic bodies, and many others (DOI: 10.1080/20013078.2018.1535750). Currently, the most relevant to the treatment of urological diseases are exosomes, microvesicles (line 483-486). Thank you so much!

Thanks very much for taking your time to review our manuscript (pharmaceutics-2447937) again. Your comments and suggestions are all valuable and meaningful for improving our paper! We have studied all reply carefully and have made conscientious correction.

Kind regards

Pu Zhang, Zhejiang Provincial People's Hospital

Reviewer 3 Report

This is a well structured and easy to read review about the application of cell membrane coated nanoparticles for tackling urological diseases. It includes a section with the main methods of preparation and functionalization, the different cell sources, and current challenges.

 Minor comments:

Line 32. Replace the coma after “circulation” by a point and followed.

Replace reference 15 by the original references.

Line 135. Define MNs.

Line 320. Define BCa instead in line 327

Line 432. Klebsiella pneumonia must be in italics. The same for S. aureus in 574 to 581.

Lines 491 and 492. Replace cm2 by cm2

Lines 492 and 493. Indicate the other types of therapy applied in that study.

In 4.6 Section, the paragraph about hybrid cell membranes could be in a separate section (4.7).

Author Response

Response to Reviewer 3 Comments

Point 1: Line 32. Replace the coma after “circulation” by a point and followed.

Response 1: Thank you for your correction. We have replaced the coma after “circulation” by a point and followed. Thank you!

Point 2: Replace reference 15 by the original references.

Response 2: Thank you for you correction. We have replaced this reference by the original references (reference 18), and updated the reference catelog in line 747-748. Thank you.

Point 3: Line 135. Define MNs.

Response 3: Thank you for your comment. “MNs” means “Fe3O4 magnetic nanoparticles”. We have modified the original text and markered it in red. Thank you!

Point 4: Line 320. Define BCa instead in line 327

Response 4: Thank you for your correction. “BCa” means “Bladder cancer”. We have defined BCa when it first appeared in line 343. Thank you!

Point 5: Line 432. Klebsiella pneumonia must be in italics. The same for S. aureus in 574 to 581.

Response 5: Thank you for your correction. We have modified the format of Klebsiella pneumonia and S. aureus in italics according to your request in line 458 and 604 to 607. Thank you!

Point 6: Lines 491 and 492. Replace cm2 by cm2

Response 6: Thank you for correction. We have replaced cm2 by cm2 in line 520 and 521. Thank you!

Point 7: Lines 492 and 493. Indicate the other types of therapy applied in that study.

Response 7: Thank you for your comment. Other types of therapy applied in that study include GNS-treated group and PBS control group (DOI: 10.7150/ntno.28450). And we have added it in the manuscript in line 523. Thank you!

Point 8: In 4.6 Section, the paragraph about hybrid cell membranes could be in a separate section (4.7).

Response 8Thank you for your comments. We have put the paragraph about hybrid cell membranes in 4.7 section and adjusted the order of some contents in 4.7 section for the reader's better reading.

Thanks very much for taking your time to review our manuscript (pharmaceutics-2447937) again. Your comments and suggestions are all valuable and meaningful for improving our paper! We have studied all reply carefully and have made conscientious correction.

Kind regards

Pu Zhang, Zhejiang Provincial People's Hospital

Reviewer 4 Report

The paper can be accepted for publication in Pharmaceutics, but before some small corrections should be made:  e.q. in the text use

Cell Membrane Coated-Nanoparticles instead

Cell Membrane Coated Nanoparticles, including the title. 

Verry small corrections should be made.

Author Response

Response to Reviewer 4 Comments

Point 1: in the text use Cell Membrane Coated-Nanoparticles instead Cell Membrane Coated Nanoparticles, including the title.

Response 1: Thank you for your kind comment. We have used “Cell Membrane Coated-Nanoparticles” instead “Cell Membrane Coated Nanoparticles”, including the title. Thank you so much!

Thanks very much for taking your time to review our manuscript (pharmaceutics-2447937) again. Your comments and suggestions are all valuable and meaningful for improving our paper! We have studied all reply carefully and have made conscientious correction.

Kind regards

Pu Zhang, Zhejiang Provincial People's Hospital

Round 2

Reviewer 1 Report

The authors addressed the concerns well.

"In 2018, Zhang and colleagues reported a groundbreaking bionanotechnology for membrane hybridization." It may be inappropriate to say this is a groundbreaking bionanotechnology.